# Fiber Optic Sensors: A Review for Glucose Measurement

**DOI:** 10.3390/bios11030061

**Published:** 2021-02-25

**Authors:** José Luis Cano Perez, Jaime Gutiérrez-Gutiérrez, Christian Perezcampos Mayoral, Eduardo L. Pérez-Campos, Maria del Socorro Pina Canseco, Lorenzo Tepech Carrillo, Laura Pérez-Campos Mayoral, Marciano Vargas Treviño, Edmundo López Apreza, Roberto Rojas Laguna

**Affiliations:** 1Doctorado in Biociencias, Facultad de Medicina y Cirugia, Universidad Autónoma “Benito Juárez” de Oaxaca. Ex Hacienda de Aguilera S/N, Calz. San Felipe del Agua, Oaxaca de Juárez 68120, Mexico; cpecmay@uabjo.mx; 2Escuela de Sistemas Biologicos e Innovacion Tecnologica, Universidad Autónoma “Benito Juárez” de Oaxaca (ESBIT-UABJO), Av. Universidad S/N, Ex-Hacienda 5 Señores, Oaxaca de Juárez 68120, Mexico; ltepech@uabjo.mx (L.T.C.); mvargas.cat@uabjo.mx (M.V.T.); elopez.cat@uabjo.mx (E.L.A.); 3Facultad de Medicina y Cirugia, Universidad Autónoma “Benito Juárez” de Oaxaca. Ex Hacienda de Aguilera S/N, Calz. San Felipe del Agua, Oaxaca de Juárez 68120, Mexico; perezcampos123@yahoo.es (E.L.P.-C.); socopina12@hotmail.com (M.d.S.P.C.); laupcm9@gmail.com (L.P.-C.M.); 4Division de Ingenierias, Campus Irapuato-Salamanca, Universidad de Guanajuato, Carretera Salamanca-Valle de Santiago km 3.5 + 1.8, Comunidad de Palo Blanco, Salamanca 36885, Mexico; rlaguna@ugto.mx

**Keywords:** glucose sensor, fiber optic sensor, glucose monitoring, biosensor

## Abstract

Diabetes mellitus is a chronic metabolic disorder, being globally one of the most deadly diseases. This disease requires continually monitoring of the body’s glucose levels. There are different types of sensors for measuring glucose, most of them invasive to the patient. Fiber optic sensors have been proven to have advantages compared to conventional sensors and they have great potential for various applications, especially in the biomedical area. Compared to other sensors, they are smaller, easy to handle, mostly non-invasive, thus leading to a lower risk of infection, high precision, well correlated and inexpensive. The objective of this review article is to compare different types of fiber optic sensors made with different experimental techniques applied to biomedicine, especially for glucose sensing. Observations are made on the way of elaboration, as well as the advantages and disadvantages that each one could have in real applications.

## 1. Introduction

Today, the measurement of glucose levels, whether in humans, animals or plants, is an activity that is carried out all the time with the aim of improving patterns, quality levels, and even leading to a better standard of living. Blood studies in medicine are commonly used for all human beings, and these results are compared and analyzed to validate the health status of each individual [1].

Diabetes is a disease that is growing at an accelerated rate throughout the world, which makes it a major health problem. The annual cost associated with managing diabetes will reach $490 billion in 2030 [2]. In 2004, the WHO estimated that the number of human beings with diabetes would increase, reaching a figure of 171 million in 2000 to 366 million in 2030 [3], while estimates from the International Diabetes Federation report 372 million diabetics in 2012 and forecast 552 million by 2030 [4,5]. It is suggested that it is necessary for people with type 1 diabetes to control their glucose two hours before and after each meal, before sleeping and at 3:00 in the morning, that is, 7 times a day: This is according to the Official Mexican Standard for the Prevention, Treatment and Control of Diabetes (NOM-015-SSA2-2010).

Commercial glucose sensors are based on puncture tests, which should be performed, as mentioned, up to 7 times a day [6]. Unfortunately, the pain, risk of infections and discomfort caused by punctures in each glucose measurement cause patients to only measure their glucose level a few times or as needed [7]. A normal person’s blood glucose concentration range is between 80 to 120 mg/dL (4.4 to 6.6 mM). A person who has more than 126 mg/dL fasting and 200 mg/dL after 2 h of eating food is considered to have hyperglycemia, and everyone who has below 54 mg/dL is considered hypoglycemic [8].

The most widely used conventional sensors for glucose control are glucometers [9], which represent a proven and mature technology that offers good sensitivity and accurate measurements in a short time. However, their follow-up method has certain disadvantages, since they require a lancet to puncture the skin and obtain a blood sample, which is then deposited on a test strip to give the result. This method makes said lancet and strip reactive once used and they must be discarded, which makes it have a substantial market for a value of $6.1 billion [10]. It is also necessary to check the expiration dates of these test strips, since after this date they may give incorrect measurement results.

Technology today can make non-invasive (NI) glucose monitoring systems [11]. Fiber optic sensors have been developed for numerous physical variables, these include thermal, mechanical and electrical magnitudes, such as temperature [12], strain [13], pressure [14], vibration [15], glucose [16], current [17] and voltage [18,19]. Fiber optic sensors can be characterized in various ways to obtain higher sensitivity, such as photonic crystal fibers [20], fiber ball [21], S-shaped fibers [22], U-shaped fibers [23], with physical modifications [24], chemical attacks [25] to the same optical fiber, so that these sensors are not disposable every time monitoring is required. Additionally one of the characteristics of these sensors is that they can be developed with the different types of fiber optics that exist.

Detecting glucose without invading or doing so by a painful means, causing some discomfort or risk to a patient is a challenge for technology, making it an important research point today. That is why Smith classified the means of detection into two groups: the minimally invasive (MI) [26,27] and the non-invasive (NI) [28,29]. Devices that use MI technology are those that need to obtain some kind of fluid or fluid from the body (blood [30,31], tears [32,33], sweat [34,35], urine [36,37,38], saliva [39,40,41], among others) in order to obtain glucose concentration, while devices using NI technology only depend on radiation without the need for a body fluid or liquid. 

Within glucose detection technologies we can find electrical methods, which take advantage of the dielectric properties of glucose at low frequencies and thus use electromagnetic radiation together with the current [42,43]. In this review, however, the objective is to discuss the work carried out for glucose measurement by means of sensors using fiber optics, a field that has made use of nanotechnology and currently the techniques of SPR and fluorescence have begun to be investigated, in combination with optical techniques.

These and many other devices have often failed to give the desired results, so they are many unmet expectations of whether it is possible to measure glucose in a NI way, since devices that use MI techniques have been used for continuous monitoring [44,45,46,47], those in the market have some disadvantages such as the need to continuously calibrate them or time of use, but with the advantage that these are not as sensitive to the need for controlled environment conditions as NI devices can be. 

## 2. Sensor Principles

### 2.1. Fiber Optic Taper Working Principle

In an optical fiber, a light wave propagates not only in the core but also in the cladding. The latter component is called an evanescent wave, which decays rapidly with increasing distance from the fiber axis, so a fraction of the incident light reflected and refracted is given by Fresnel’s coefficients and depends on the angle of incidence and refractive indices of the core and cladding [48]. Therefore, when an optical fiber is tapered both the cladding and core diameter are reduced in size, and this increases the intensity output of light waves. When the light passes through the tapers lower and higher order modes are generated due to the core and cladding size [16]. The higher order modes are generated easily formed in tapers in form of evanescent waves (EW) due to the fact the core and cladding medium are smaller, see Figure 1 [21,49]. The EWs attenuate to 1/e during propagation distance into taper, which is known as the penetration depth [16,50]:(1)$$d_{p}=\frac{\lambda }{2\pi \sqrt{n_{co}^{2}\;\sin\theta_{t}^{2}-\;n_{cl}^{2}}}$$
where *λ* is the wavelength of the light, *θ* is the angle of incidence at the interface between the core and the cladding, and $n_{co}$ and $n_{cl}$ are the refractive indices of the core and cladding, respectively. 

The number of modes that can propagate into fiber is related to the normalized frequency *V* value [16]:(2)$$V=\frac{2\pi \alpha }{\lambda }\sqrt{n_{co}^{2}-n_{cl}^{2}}$$
where *α* is the radius of the core. The condition for the single-mode fiber is *V* < 2.405 for only the fundamental mode. However, in the tapered fiber transition region various modes can exist at the same time, because the value of *V* depends on the excitation of the modes, due to the difference in the refractive indices, therefore, the *V* value of the cladding increases beyond the cutoff value as [16]:(3)$$V_{cl}\left(z\right)=\frac{2\pi \alpha \prime \left(z\right)}{\lambda }\sqrt{n_{cl}^{2}-n_{ex}^{2}}$$
where $n_{ex}$ represents the refractive index of the external medium.

The tapers allow the coupling between the cores and cladding modes when a broad light source propagates into the tapered region zone. There are adiabatic and non-adiabatic taper structures: A tapered fiber will be adiabatic when the transmission loss is almost negligible due to the fact that the energy maximum remains in the fundamental mode $HE_{11}$ (it can go up to 99.5%) and on little energy goes into the excited high-order modes [51]. However, when the taper angle is large enough, the coupling length between the fundamental and higher order cladding mode is said to be non-adiabatically tapered. Therefore, the total transmission is due to the fundamental mode $HE_{11}$ and the high order modes $HE_{12}$ are stronger [52,53]. 

Now, the depth of penetration at the interface of the lining and the surrounding medium can be calculated as follows [54]:(4)$$d_{pl}=\frac{\lambda }{2\pi \sqrt{n_{cl}^{2}\sin\;\theta_{i}\;-n_{ex}^{2}}}$$

### 2.2. U-Shaped Fiber Optic Working Principle

As well known, optical fibers have several distinct advantages such as low cost, small size, flexibility and remote sensing opportunities. When there is at least one degree in the radius of curvature in an optical fiber it is possible to observe interference phenomena in the spectrum. When the light source hits the part of the curve in the optical fiber, a light leakage is generated because the core layer of the fiber and the interface of the cladding layer of the curved part already do not maintain total reflection, as shown in Figure 2 [23].

U-shape bending is generally done on uniform fibers so that core radius decreases, which results in the propagation of high order modes. The light is launched from one end of the fiber and is detected at the other end. The transmitted power depends of the loss of the evanescent field penetration in the bending region. However, there is some lights are reflected on the cladding-air interface, giving rise to the radiation. The reflected light in the cladding is known as the whispering gallery mode (WGM) [55]. The Mach-Zehnder interferometer model can be used to calculate the output interference pattern due to the coupling of the WGM and core mode [56]. Therefore, the interference spectrum in a U-shaped fiber optic light intensity sensor is [57]:(5)$$I=I_{co}+I_{wis}+2\sqrt{I_{co}+\;I_{wis}}\;\cos\;\left(\phi \right)\;$$
where $I_{co}$ and $I_{wis}$ are the light energy intensity of the core mode and the WGM, respectively, and ϕ is the phase difference between the cladding mode and WGM. The expression is given [53,56,57]:(6)$$\phi =N\left[\frac{2\pi }{\lambda }\left(n_{cl}2L-\;n_{eff,co}Z\right)+\phi_{r}\right]\;=\left(2k+1\right)\pi$$
where *N* is the number of reflections of the WGM on the cladding-air interface at the bending region, *L* is half of the path length between two reflection points, and $n_{eff,co}$ is the effective refractive indices of the core mode, *Z* is the arc length of the core, $n_{cl}$ are the refractive index core base mode and m level cladding mode, $\phi_{r}$ is the change in phase at the cladding-air interface and k is an integer [58].

The transmission wavelength dip at which destructive interference occurs at *k*th order of the resonance WGM, and can be expressed as [15,59,60]:(7)$$\lambda_{res,WGM}=\;\frac{2\Delta n_{eff}R}{2k+1}$$
where $\Delta n_{eff}$ is the difference of the effective refractive indices between the core mode and *k*th order cladding mode. Therefore, the transmission spectrum produces multiple resonance WGM wavelengths [61].

### 2.3. Surface Plasmon Resonance (SPR)

Surface plasmon resonance happens when the interface of a highly conductive metal and dielectric material are excited by an electromagnetic field, and this resulting field decays exponentially, making it is very sensitive to changes in the refractive index of the surrounding environment [62].

The propagation constant field of a surface plasmon $\beta_{sp}$ propagating along a planar boundary between a semi-infinite metal with a complex permittivity $\varepsilon_{m}=\varepsilon_{m}^{\prime }+i\varepsilon_{m}^{\prime \prime }$ and a semi-infinite dielectric with a refractive index n is expressed by [63]:(8)$$\beta_{sp}=\frac{\omega }{c}\sqrt{\frac{n^{2}\varepsilon_{m}}{n^{2}+\varepsilon_{m}}}$$
where ω is the angular frequency and c is the speed of light in the vacuum.

The schematics of an SPR optical sensor are shown in Figure 3. The experimental setup consists of a light source, an SPR coupler and a detector. The lightbeam is introduced to the SPR coupler to excite a surface plasmon, and then, the diffracted or reflected light is detected by the detector. As a change in the refractive index at the metal-dielectric interface results in a change in the effective index of the surface plasmon, therefore any change in the refractive index can be measured when changes the intensity, angular or wavelength of emission spectrum of surface plasmon occur [64].

In a taper fiber sensor, the refractive index of the core and the cladding are modified. This helps the variations that can be obtained in glucose levels be characterized as changes in the refractive index corresponding to the change in resonant frequency. Thischange is the so-called SPR [66], and is how a displacement in the loss of reflection intensity is obtained. SPR can be combined with some other techniques as shown in Figure 4 applied on a tapered optical fiber and gold film.

### 2.4. Fiber Bragg Grating (FBG)

An optical fiber Bragg grating (FBG) is a one-dimensional periodic modulation of the refractive index which is normally formed based on the fiber photosensitivity. The Bragg grating are made by illuminating the core of a suitable optical fiber a spatially varying pattern of intense ultraviolet (UV) laser light. A periodic spatial variation in the intensity of UV light, caused by the interference of two beams placed over the fiber, gives rise to a corresponding periodic variation in the refractive index of the fiber [67].

Figure 5 shows a schematic diagram of a FBG, and its effect on the incident light. When the reflected light has a wavelength equal to the Bragg wavelength it is reflected back to the input while other wavelengths are transmitted [67,68]:(9)$$\lambda_{B}=2n_{eff}\Lambda ,$$
where $\lambda_{B}$ is the Bragg wavelength, $n_{eff}$ is the effective index and Λ is the grating period. The Λ is constant over the length of the grating, therefore, a grating is a device that periodically modifies the phase or intensity of wave reflected or transmitted through it.

There are reported works on fiber optic sensors for glucose detection developed using the Fiber Bragg Gratings (FBG) technique. Examples of these sensors are those that are placed in the radial artery to measure the voltage (pulse 302 wave) [68], here, to detect concentrations of 0 to 60% they used doping in the gratings of various substances including nitric acid [70,71] and aminophenylboronic with a detection range of 1 mM to 10 mM [72]. In another work a Tilted Fiber Bragg Grating (TFBG) was used in a glucose sensor. In [73] a silver-plated TFBG sensor was used to detect blood glucose at concentrations of 0 mM to 12 mM, while in [74], the authors doped a sensor with polydopamine with a detection limit of 10^−7^ M, or with graphene oxide and glucose oxidase in glucose ranges from 0 to 8 mM which achieves a response coefficient of 0.24 nm/mM, see Figure 6 [75].

## 3. Sensors

### 3.1. Multimode Fiber Optic Sensors

In 2010 Saxl used a multimode fiber with a 1000 µm core (Thorlabs, Ely, UK), with which he formed a 2 mm long chamber glucose sensitive using agarose or polystyrene beads with a section of a glass tube of 1.5 mm in diameter and secured it with epoxy resin on the tip of the optical fiber as shown in Figure 7. The sensor uses a 417 nm laser and obtainsg results at 542 nm, with this they obtained a response time of 1 h [76], while two years later Singh obtained results in 60 s, in a range of 600 nm to 750 nm, varying these results based on the sensed glucose concentration (Figure 8) with a sensor design elaborated on the basis of a multimode optical fiber with a diameter of 600 mm at the core and 0.4 apertures, with a 20 cm fiber length [77].

In 2015 Li used a multimode fiber with 600 µm core, where said fiber is doped with a 5 nm chromium layer and a 50 nm gold film to obtain greater sensitivity, which was used with glucose samples ranging from 1 mg/dL to 300 mg/dL, with results detected from 625 nm up to 668 nm, according to the samples [78].

Milenko in 2018 used a multimode fiber optic with a 105 mm core and 125 mm cladding diameter. Light coupling and Raman signal collection were carried out through one end of the fiber, while the other end was immersed in water-glucose solutions with a concentration range between 0–1110 mM. Measurements were taken with a scan time of 10 s and 20 s acquisitions using an laser excitation wavelength of 785 nm [79].

### 3.2. Plastic Fiber Optic Sensors

In 1989, Trenttnak developed a glucose biosensor based on intrinsic green fluorescence, working at the 450 nm range in which plastic optical fibers work, and where glucose fluorescence is excited [80]. Later, in 2008 Binu performed experiments to detect refractive indices while varying the concentration of glucose in distilled water. This sensor consists of a plastic fiber optic probe, a reflecting mirror, a temperature controller, a photodiode detector and a digital multimeter. This sensor is shown in Figure 9 [81].

In 2015 Qiu designed a plastic fiber optic sensor, with a light source, polarization device, reaction zone, spectrometer (USB2000+ miniature fiber optic spectrometer) and a PC. Qui used a LED light source to generate red light of 628 nm wavelength, making measurements on samples of 1–40% glucose concentration [82]. This same year, Verma developed an experimental setup for the characterization of urea and glucose levels (channel 1 and channel 2) shown in Figure 10, having a detection range between 500 to 800 nm with glucose samples on concentrations from 0 to 260 mg/dL. In this sensor, which has two narrowings (one in each channel), he used copper and tin oxide as dopants, respectively, to improve the sensitivity [83].

Tiangco in 2016 used a system consisting of 15 inches of single core plastic fiber optic with a diameter of 2 mm, a core of 1.96 mm and an outer diameter of 3 mm (Edmund Optics, Barrington, NJ, USA), working in a visible light range from 366 to 540 nm, with glucose samples of 4–20 mM and using a NI-NTA agarose beads structure dopant to have a sensitivity of 2.888 ± 0.08085 [84]. The same year Yunianto created a sensor to detect the content of glucose in blood through a plastic optical fiber, based on a green LED. The test is carried out in a UV-VIS spectrometer which gives a result in 581 nm with samples of 81.2–235.1 mg/dL [85]. Zhao manufactured a sensor which uses acetic acid under an ultrasonic power of 130 W and a temperature of 25 °C, presenting a high sensitivity of 9.10 RIU g/L^−1^ for glucose solutions between 10% and 80% concentration, obtaining results in the range of 500 to 750 nm [86].

Azkune in 2018 recorded and compared different glucose absorption spectra in plastic optical fibers doped with methyl methacrylate. These tests are performed by immersing the sensor for 10 min in different glucose concentrations ranging from 0 to 25 mM [87]. In 2019, normalized gamma spectra in a visible range of 450 to 500 nm for different glucose concentrations were located, showing a change in intensity and a shift to the right [88].

### 3.3. Photonic Fiber Optic Sensors

Asher in 2003 developed a sensor that shows a symmetric diffraction peak at 496 nm, which indicates that it diffracts blue-green light. This diffraction peak shifts towards the red as the glucose concentration increases; the sensor diffracts green light at 506 nm for 1 mM glucose, orange light at 576 nm for 20 mM glucose, and red light at 624 nm for glucose, as shown in Figure 11 [89].

In 2016, Mohamed suggested a sensor that offers a high sensitivity at 422 nm/RIU with high linearity, obtaining results on glucose concentrations of 1400 to 1420 g/L [90]. Mohammad presented and discussed the design of a surface plasmon photonic glass fiber (PCF) biosensor for glucose monitoring. The numerical results show that the D-shaped plasmonic PCF after etching three rows of air holes and using gold as a doping structure can reach a sensitivity of 200 nm/RIU [91].

In 2018 Natesan proposed a blood glucose sensor using a trinuclear photonic glass fiber, where a hollow channel filled by analyte acts as a liquid core and two nuclei remain considering the silicon substrate having a sensitivity of 23267.33 nm/RIU. The transmission spectrum shows a maximum wavelength shift for polarization at 470 nm [92]. This same year, Hossain presented an optimized hexagonal photonic glass fiber (PCF) geometry for the detection of glucose concentrations of 20%, 30%, 40%, 50% and 60% in water at wavelengths ranging from 1200 nm to 1600 nm [93].

### 3.4. Single-Mode Fiber Optic Sensors

In 1996 Rosenzweig tested a single-mode fiber with a few micrometer cores, placing said fibers in chloride for 10 s, removing about 3 to 5 cm of the coating, leaving a tip heated at a power of 10 mW with a laser beam, to form a glucose sensor with a reaction time of 1.5 s [94]. Later, in 2010, Shao developed a configuration for a quartz-coated silicon single-mode fiber optic LSRP sensor (P600-4 UV-VIS, Ocean Optics Co.). Shao removed a 2 cm length of the coating layer iusing a solution of HF as shown in Figure 12, and usedgold as a doping agent to obtain a sensitivity of 13.09 nm/RIU [25].

In 2016 Yin manufactured a sensor using a single mode fiber (RC1550 8021/165 Yangtze Optical Fiber and Cable Joint Stock Company Limited, Wuhan, China) with a diameter of 80 µm. Initially the fiber was charged with hydrogen at a pressure of 1500 psi for a week, and after doping said fiber with glucose oxidase, it achieves a sensitivity of 205 nm/RIU for glucose concentrations of 2 µM to 10 µM [95]. This same year, Fang removed 60 mm of protective layer from a single-mode optical fiber inside a 300 mm long fiber, fixed the ends to a mobile platform to create an angle in said fiber, and tested of 6 to 30% glucose concentrations achieving a maximal sensitivity of 0.85 dB/% and linear dependence coefficient was 0.925 [96].

Sun in 2017 used a single-mode fiber optics sensor and experimental setup of a light source, a spectrum analyzer (OSA, AQ6370B) (Yokogawa Electric, Tokyo, Japan), a circulator and a test zone, and a superluminiscent LED with a length of 1310 nm wave in samples from 0 to 60 mM of glucose. Figure 13 shows the results obtained with the spectrum analyzer and graphed using the Origin software [97]. The same year, Gandhi created a single-mode fiber optic biosensor, which reaches 12820 nm/RIU with a resolution of 7.8 × 10^−3^ RIU, demonstrating that this is a highly sensitive refractive index sensor for glucose detection applications [98].

In 2018 Novais proposed a sensor consisting of a single-mode fiber section spliced to a short section of coreless silicon optical fiber. By doping with hydrofluoric acid, he obtained a sensitivity of 1467.59 nm/RIU for the shifts in the refraction of different percentages of glucose concentration ranging from 0 to 60% [99].

Khan manufactured an optical fiber coated with gold nanoparticles; he selected a single mode fiber 70 mm long, with core and cladding diameters of approximately 3 µm and 125 µm, respectively. Later 10 mm of cladding was removed from one end, thereby creating a tip, which is cleaned with ethanol, methanol and deionized water and then dried with N_2_ gas, obtaining results at 820 to 920 nm and having a response time of 8 to 9 s in each measurement, with a sensitivity of 3.25 nm/mM [100]. Chen proposed a U-shaped fiber optic sensor made using a flame heating method. This was packaged in a glass tube to reduce signal loss during tests as a glucose sensor demonstrating that when one immerses the sensor in different concentrations of glucose ranging from 0 to 8%, the wavelength or transmission changes as shown in Figure 14 [23].

In 2020, Yang presented a study of a glucose sensor based on a single-mode fiber optic structure using graphene oxide and gold nanoparticles. The optical fiber is tapered with a 25 µm waist and a 6 mm tapered region. This conical region is covered with graphene and gold nanostructures to increase biocompatibility. It shows that good absorbance properties at wavelengths of 230 nm and 519 nm with a sensitivity of 1.06 nm/mM Figure 15 [16].

## 4. Discussion

Diabetes continues to be a global health problem, and at least for the moment, without a cure, so it is important for every patient with this disease to keep their glucose level monitored within a healthy range to continue enjoying a relatively healthy life. Glucometers today are the most reliable devices for taking daily glucose measurements, but technology advances day by day and that has led us to create glucose sensors through fiber optics. This will avoid punctures and pain for the patient to obtain a blood sample and it will also help to avoid having to continuously calibrate the sensor or consume lancets that are only for a single use.

For glucose measurement, there have also been many technological advancesSmith in his book [101] thoroughly summarizes past and current instruments used in non-invasive glucose (NI) and minimally invasive (MI) monitoring. Some devices after going on sale were withdrawn (GlucoWatch, Pendra, among others), due to their problems of accuracy in results and time of use, and other projects such as Google and Novartis lenses were also ruled out by the difficulty of accurately obtaining glucose in tears due to the small amount of glucose in them and sample collection, and along with this technology other devices based on other techniques have been developed, including optical polarimetry [102,103], electrode detection [104], by fluorescence with devices developed for the control of glucose under the skin such as the Senseonics Eversense^®®^ (with 180 days of service life) or the new device developed by Profusa, as well as the Dermal Abyss tattoo of, which considers the surface of the body as an interactive display, producing color changes in response to glucose concentration [105], however, the fact remains that this is still invasive, at least for the first time.

In fluorescence glucose measurements the intensity or time of signal drop can be detected. The lifespan of fluorescence is unique for each analyte which can be measured in dispersion media [106], this helps for the difference between each substance [107]. In spectroscopy water does not absorb much radiation, so light can pass through the stratum corneum of the epidermis to achieve a higher blood concentration, regardless of skin color [108], making it the first choice to develop technology for NI glucose sensors. As shown in Table 1, it also has disadvantages and interference from proteins and acids, which make the glucose-like absorption decrease affecting the reliability and sensitivity to glucose [109]. In contrast, the technology developed based on electromagnetism measures glucose by current or voltage according to magnetic coupling [110]. This operation depends on the relationship between the input and output voltage or amperage, which is proportional to the glucose concentration.

Glucometers today have many advantages, as the results are delivered in seconds, but it cannot be considered to be the best way to monitor glucose, this because it is painful for patients to be pricking the tip of the finger each time for each sample [111], to this one must add the fact that the test strips used by glucometers that are currently on the market are single use and susceptible to electromagnetic interference. Proposals to monitor glucose using elaborate fiber optic sensors can be used as many times as necessary, but with the disadvantage that after a certain period calibration is needed, and they are sensitive to the ambient temperature where glucose monitoring is being done [112].

In Table 2, we list see some characteristics of sensors currently in their prototype stage. All these optical methods share some characteristics, but all use a different technique or type of sensor. They work in different detection ranges, even when using the same doping structure, in this case gold, and it is possible to see that photonic fibers work between 1663–1665 nm [90] and single mode fibers from 500 to 550 nm [25]. In this case the photonic fiber uses a laser as a source, while single-mode fibers work with a visible light source. A plastic fiber optic sensor which is doped with methyl methacrylate was also designed to operate at 450–500 nm [88] as well as a single-mode fiber optic working at 488 nm [94] but it is doped with methylene chloride, both being able to work in visible light.

Response times are becoming important for glucose measurements. Since some patients take samples up to 7 times a day in it is necessary that the sensor does not take long between each measurement. There are biosensors that deliver results in only a few seconds [77,82,85,97] but the sensitivity of fiber-optic-based sensors may have a margin of error [88,90]. With these two parameters, we can say that they can surpass the glucometers that we commonly use. Fiber optic sensors for glucose measurement can not only improve the measurement parameters of conventional glucometers, they can also measure other parameters [83].

Based on Table 2 some parameters and characteristics of each developed sensor are shown, where each sensor was characterized according to the instruments used, from the optical fiber used, the type of source, light color, technique used, in reflection or transmission, if the technique was used in the extreme or medium area and type of spectrometer. All these characteristics are important to be able to achieve the result obtained both in sensitivity and in detection range. There are also many other parameters, which are part of this characterization, but not all were included since not all the articles consulted contain all the complete data and to maintain the uniformity of the parameters in Table 2 we decided to omit them for the comparison.

## 5. Conclusions

The technological advances achieved in glucose monitoring through fiber optics have continued to advance and improve, although all these optical technologies have advantages and disadvantages. Optical fibers have many properties which can be modified to obtain better results in both detection and sensitivity. The fibers can be narrowed, doped, bent or some physical change made, both in their core and in their cladding.

Technology in recent years has seen surprising growth, and carrying devices as a garment is common today, with devices in the form of bracelets, earrings, watches, rings, necklaces, among others. Thanks to this “wearable” technology it is possible to record many physiological parameters, which previously could only be obtained with larger devices or going to doctor. Some examples are technologies used to measure blood pressure, control heart rate, count steps, and oxygenation in the blood, among others. With all these advances, it is possible to collect data to control and monitor a disease until one can notify a doctor of any symptoms that you may present immediately.

Table 2 shows the comparison of some characteristics of the sensors made with four different types of optical fibers, all showing results positive, some with higher sensitivity at different response times. This leads us to believe in the possibility of seeing these new types of glucometers within the reach of every patient in a short time, with the advantage of not needing punctures or causing any pain to the patient, while being economically more accessible and less harmful to the environment.

## Figures and Tables

**Figure 1 biosensors-11-00061-f001:**
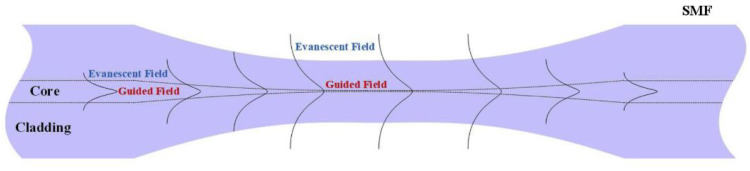
Propagation of light through a narrowing [16].

**Figure 2 biosensors-11-00061-f002:**
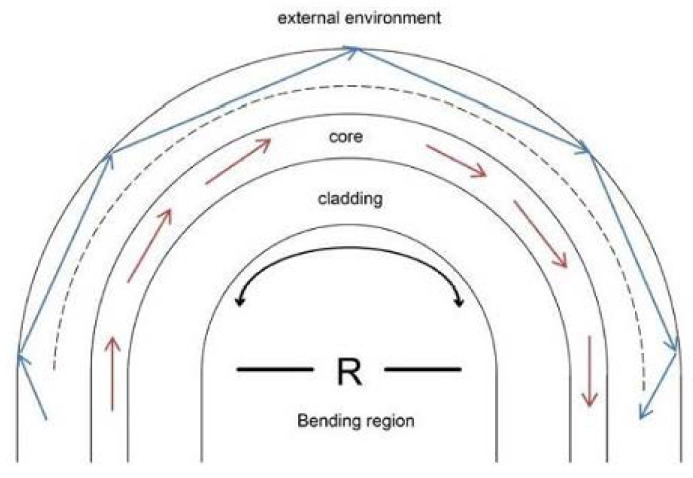
U-shaped fiber optic schematic diagram [23].

**Figure 3 biosensors-11-00061-f003:**
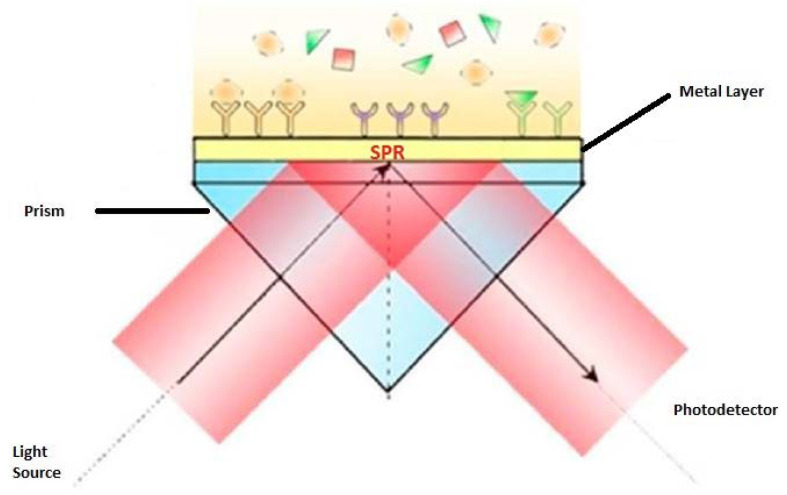
Configuration of SPR excitation [65].

**Figure 4 biosensors-11-00061-f004:**
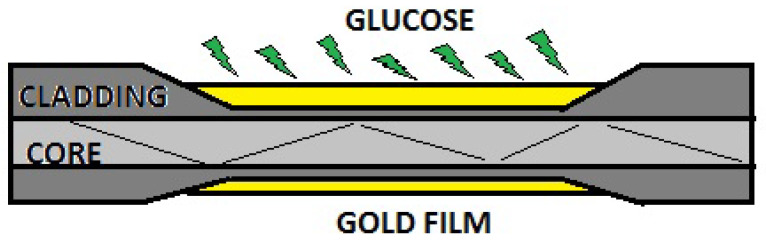
Tapered plasmonic fiber optic coated with gold film for increased sensitivity in the region tapered waist.

**Figure 5 biosensors-11-00061-f005:**
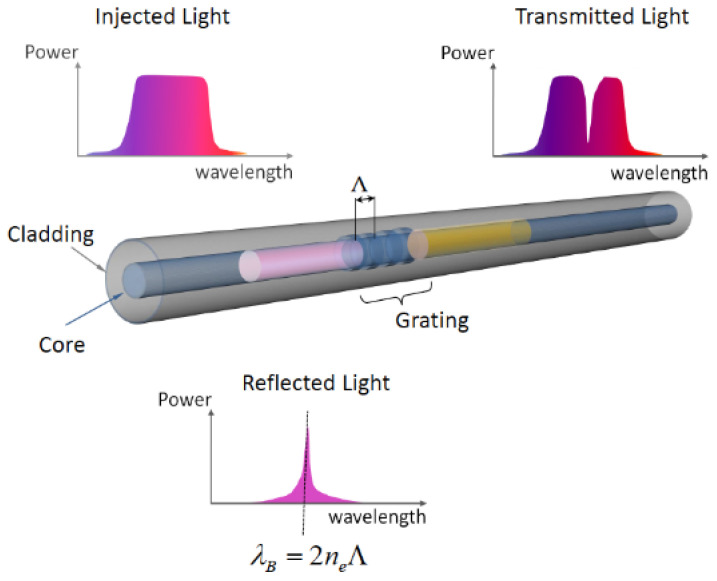
Diagram and working principle of fiber Bragg grating [69].

**Figure 6 biosensors-11-00061-f006:**
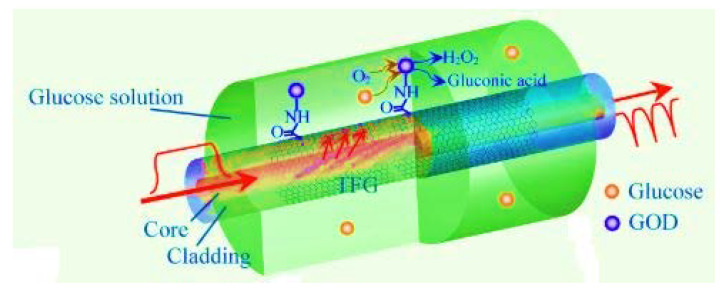
Schematic of the sensor for glucose detection in TFBG coated [75].

**Figure 7 biosensors-11-00061-f007:**
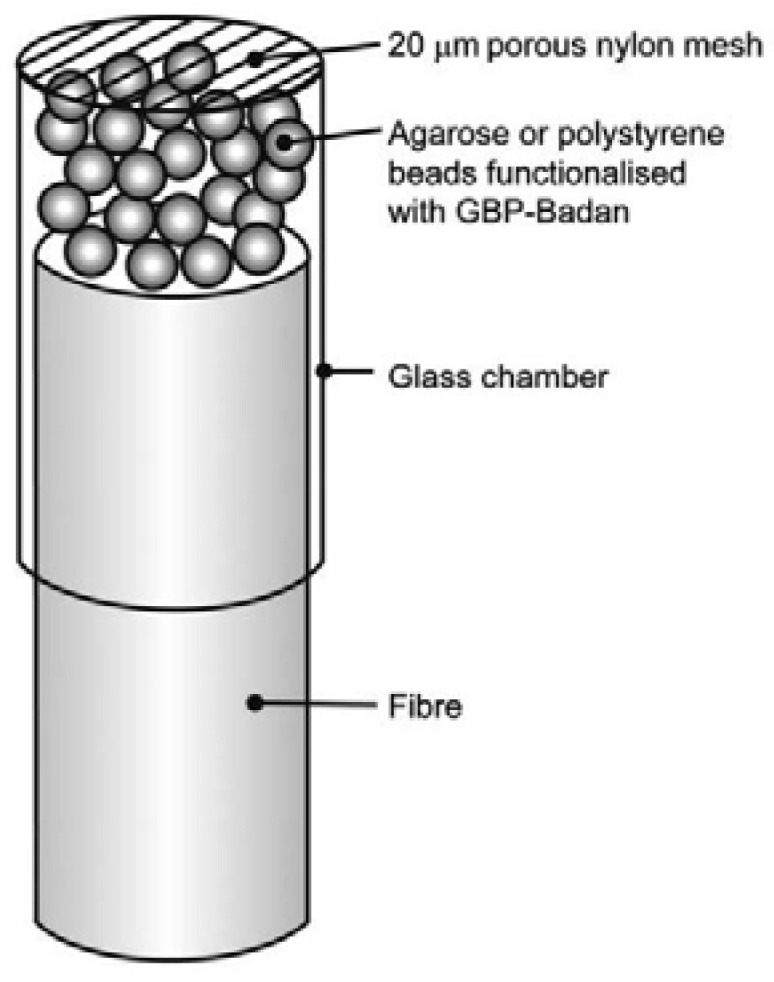
Multi-mode fiber optic sensor tip with beads [76].

**Figure 8 biosensors-11-00061-f008:**
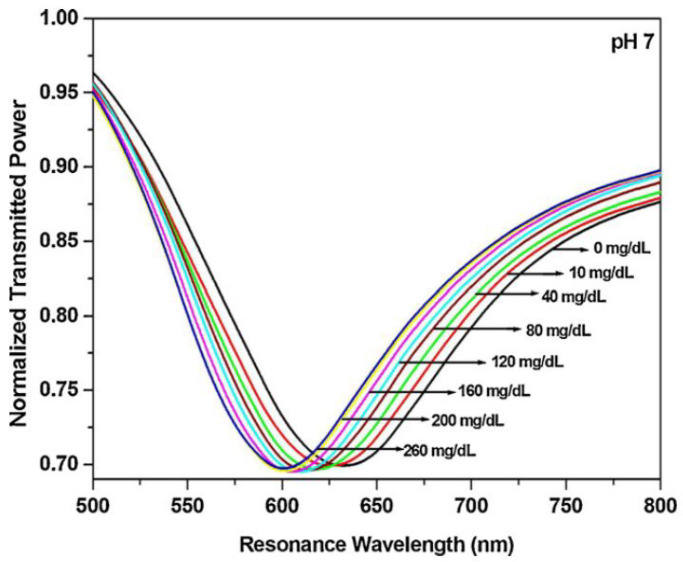
Optical sensor spectrum with different glucose concentrations in a range of 0–260 mg/dL [77].

**Figure 9 biosensors-11-00061-f009:**
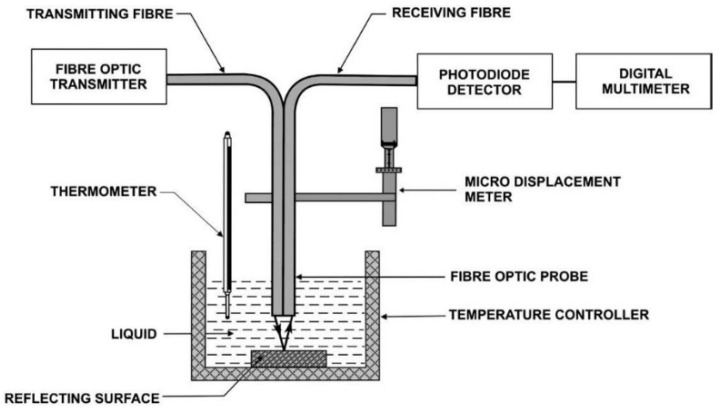
Schematic experimental setup to detect variation in the refractive index based on glucose concentration [81].

**Figure 10 biosensors-11-00061-f010:**
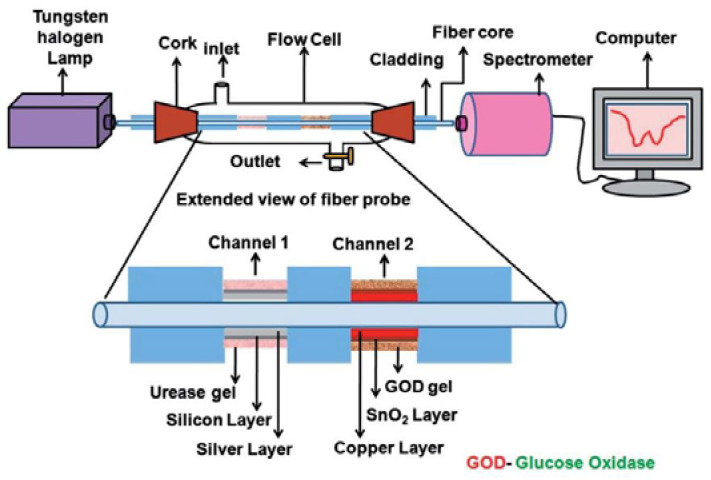
Experimental diagram of urea and glucose sensor [83].

**Figure 11 biosensors-11-00061-f011:**
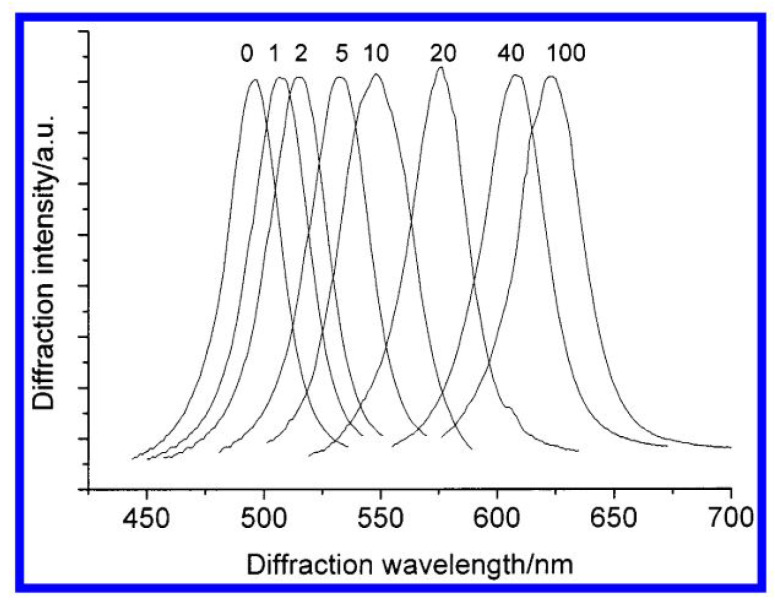
Glucose concentrations from 0 to 10 mM [89].

**Figure 12 biosensors-11-00061-f012:**
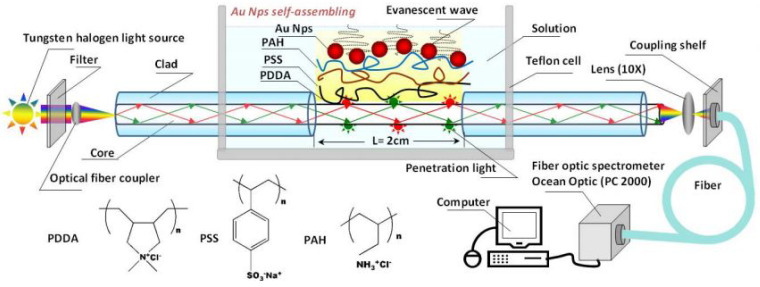
Schematic drawing of the optical fiber in different stages [25].

**Figure 13 biosensors-11-00061-f013:**
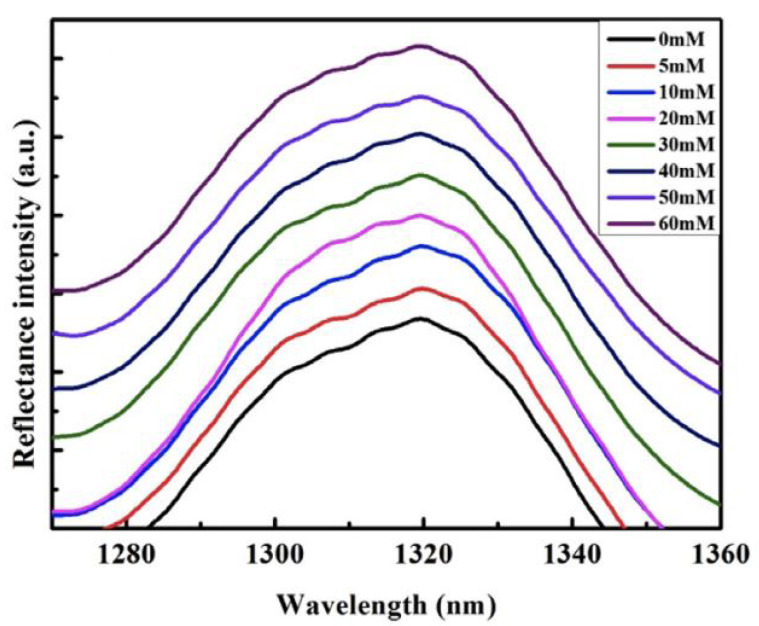
Spectra of glucose concentrations from 0 to 60 mM [97].

**Figure 14 biosensors-11-00061-f014:**
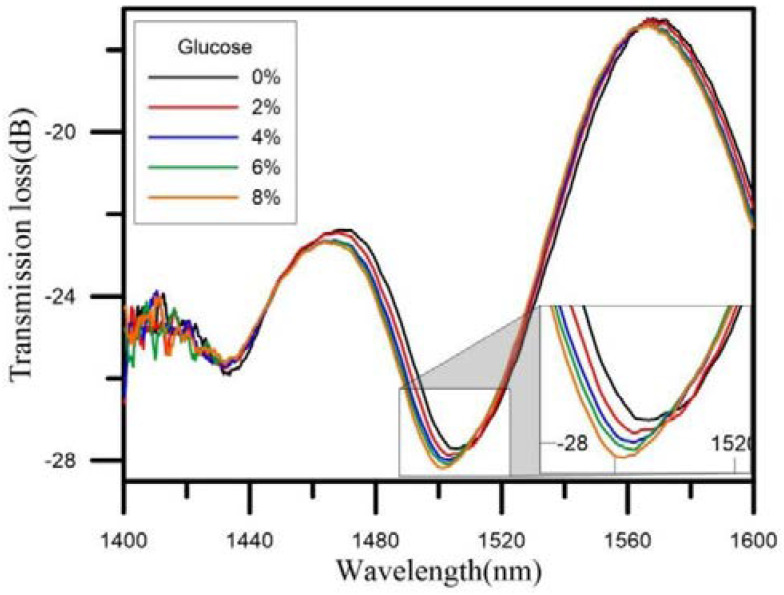
Wavelength change at concentrations of 0–8% glucose in water [23].

**Figure 15 biosensors-11-00061-f015:**
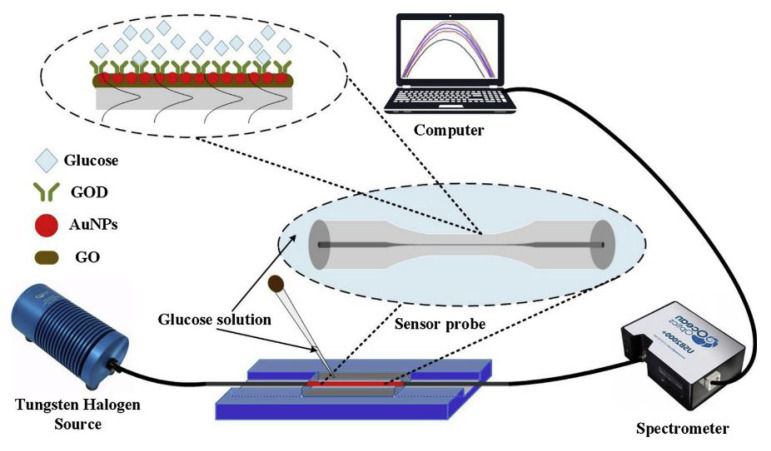
Experimental arrangement of the glucose sensor described in [16].

**Table 1 biosensors-11-00061-t001:** Advantages and disadvantages of the optical fiber sensor.

Sensor Fibers Technology	Advantages	Disadvantages
Fluorescence	Immune to light spreading of tissue	Short useful life
	High specificity	Toxicity problems
	Sensitive to very small concentrations of glucose	Noticeable to interference
Spectroscopy	Absorption band specifies	Fake readings
	High specificity	Strong water absorption
		Low level of penetration
		Temperature-susceptible
Electromagnetic	Single-frequency use	Very sensitive to temperature
	No risk of ionization	Sensitive to electromagnetic fields
SPR	They are not susceptible to electromagnetic interference	Calibration process
	They’re not disposable.	Temperature sensitive
	Very sensitive to small changes in glucose	Sensitive to movement
	Instant results	Bulky in size
	Minimally invasive	
	Continuous monitoring	

**Table 2 biosensors-11-00061-t002:** Comparison of glucose sensors made with fiber optics.

Type of Fiber	Parameters						Ref
	Sensing System	Detection Range	Dynamic Range	Immobilization Structure	Response Time	Sensitivity	
MMF	In the extreme	N/A	45–360 mg/dL	Boronic acid with fluorescent dye	5 min	N/A	[113]
	In the extreme	542 nm	0–100 mM	Agarose or polystyrene	1 h	N/A	[76]
	Middle area	600–750 nm	0–260 mg/dL	Silver and Sicilian	60 s	N/A	[77]
	Middle area	N/A	N/A	gold	N/A	3632 nm/RIU	[114]
	Middle area	625–668 nm	1–300 mg/dL	Borate chromium gold polymer	N/A	N/A	[78]
	N/A	625–700 nm	0–500 nm/dL	Glucose oxidase and polyacrylamide	22 s	0.14 nm/(mg/dL)	[115]
	In the extreme	N 785 nm	0–1110 mM	N/A	10–20 s	N/A	[79]
PCF	N/A	496–624 nm	0–100 mM	Boronic acid	N/A	N/A	[89]
	N/A	1400–1420 nm	30–330 g/L	N/A	N/A	422 nm/RIU	[90]
	In extreme	600–750 nm	40–400 mg/dL	N/A	N/A	N/A	[59]
	N/A	1663–1665 nm	N/A	gold	N/A	200 nm/RIU	[91]
	N/A	440–470 nm	10–20 g/L	Silicon substrate	N/A	23,267.33 nm/RIU	[92]
	N/A	1200–1600 nm	20–60 %	N/A	N/A	N/A	[93]
	N/A	4950–6930 nm	10–40 g/L	N/A	N/A	6930.6 nm/RIU	[20]
POF	In extreme	450–500 nm	1.5–2 mM	Glucose oxidase	2–30 min	N/A	[80]
	In extreme	660 nm	0–25 g/dL	N/A	N/A	0.0072 V/wt	[81]
	Middle area	620–635 nm	1–40%	Graphene	N/A	N/A	[82]
	Middle area	500–800 nm	0–260 mg/dL	copper and tin oxide	N/A	N/A	[83]
	In extreme	366–540 nm	4–20 mM	NI-NTA Agarose Beads	50 s	2.888 ± 0.08085	[84]
	Middle area	581 nm	81.2–235.1 mg/dL	N/A	N/A	N/A	[85]
	Middle area	500–750 nm	10–80%	Acetic acid	N/A	9.10 [(RIU)(g/L)]^−1^	[86]
	In extreme	N/A	0–25 mM	methyl methacrylate	10 s	N/A	[87]
	Middle area	450–500 nm	N/A	Methyl methacrylate	N/A	N/A	[88]
SMF	In extreme	488 nm	0.7–10 mM	Methylene chloride	1.5 s	N/A	[94]
	Middle area	500–550 nm	N/A	gold	N/A	13.09 AU/RIU	[25]
	Middle area	1510–1520 nm	2–10 µM	Glucose oxidase	6 min–70 s	205 nm/RIU	[95]
	Middle area	1490–1532 nm	6–30%	N/A	N/A	0.85 dB	[96]
	In extreme	1545–1560 nm	0–60%	Hydrofluoridric acid	N/A	1467.59 nm/RIU	[99]
	In extreme	1280–1340 nm	0–60 mM	Hydrofluoridric acid	N/A	0.1787%/nM	[97]
	Middle area	880 nm	10–50%	Gold	N/A	12820 nm/RIU	[98]
	In extreme	820–920 nm	1 µM–1 M	Gold	8–9 s	3.25 nm/mM	[100]
	Middle area	1480–1520 nm	0–8%	Gold and Glucose oxidase	N/A	5.101 dB/%	[23]
	Middle area	230–519 nm	0–11 mM	Graphene Oxide and Gold	N/A	1.06 nm/mM	[16]

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
