# Peer review of "Fiber Optic Sensors: A Review for Glucose Measurement"

_biosensors, 2021, doi:10.3390/bios11030061_

Round 1

Reviewer 1 Report

This manuscript presents a short review on the technological advance in fiber optics SPR sensor for glucose monitoring. This manuscript is very interesting and benefit for the fundamental and applied study of fiber optics-based SPR sensor. Thus, I recommend the publication of this work in Biosensors, after the following minor points have been considered by the authors:

  1. The comparison (advantage and disadvantage) of fiber optics-based methods with other existing methods should be discussion.
  2. Recently, Au-enhanced fiber optics SPR sensors for glucose detection have been proposed by others, such as Peng et al. (Biosensors and Bioelectronics, 2018, 117, 637), Jit et al. (nature scientific reports, DOI: 10.1038/s41598-020-67986-4), and so on. Considering the integrity of this review, these new technologies should be introduced in this review.
  3. The existence question and development tendency of fiber optics-based SPR sensor for glucose detection should be discussed in more details.

Author Response

We highly appreciate the time and suggestions from reviewers, which were addressed. Please find the changes in the manuscript marked with red font, and in the PDF file our responses.

Reviewer 2 Report

In this review, the authors discuss fiber optic sensors for glucose measurement. In general, glucose monitoring is an extensively studied subject and a wide variety of technologies have been developed through recent years to address its simple, fast, and reliable detection.

In general, this revision is highly descriptive without any comparison between the studies presented by the authors and, in a few bits, a little bit confusing. Nonetheless, the authors did a good job organizing and discussing the general sensor principles and added a table to compare the glucose sensors described in the text.

However, this manuscript needs major editing improvements and reorganization before consideration for publication. For instance:

1) The manuscript is too descriptive and information about the LODs, linear ranges and aspects on how the glucose detection is performed should be added to the manuscript. For instance, only a few studies seem to have tested the sensors in blood samples and these are mentioned in the text by the authors. But are the remaining glucose fiber optic detection systems used in blood samples? Furthermore, is blood the only sample where glucose can be monitored?

Are any of these sensors capable of transdermal glucose monitoring?

2) In the first paragraph of the introduction, the authors state the measure of the levels are important. What levels are important? Also, this paragraph lacks references.

3) In 2017 FDA approved a continuous blood sugar monitor for diabetes that doesn’t need backup finger prick tests. This technology uses a glucose sensor and should be referenced in your manuscript with a brief discussion about its characteristics and how fiber optic glucose sensors can outperform these sensors. (https://www.abbott.com/corpnewsroom/diabetes-care/revolutionizing-cgm-with-freestyle-libre.html)

4) In general, a continuous blood sugar sensor should be the standard for comparison with the fibber optic studies presented in this review. In fact, glucose sensors are a widespread subject in the literature, and the reason why fiber optic sensors are a relevant subject should be better explained in the manuscript. Furthermore, these continuous monitoring systems seems to be the trend in glucose monitoring for diabetes patients. This is stressed when these are also reaching the market and making an impact with important references such as a) and b), which are missing in this manuscript.

a) Müller et al., J. Diabetes Sci. Technol., 2013; 7: 13-23. DOI:10.1177/193229681300700103 b) Mohamed Elsherif et al., Biosens. Bioelectron, 2019; 137: 25-32. DOI: 10.1016/j.bios.2019.05.002

For this reason, the authors should add a section with the most recent developments about fiber optic continuous glucose monitoring systems in this work.

5) Is there any example capable of monitor glucose in a non-invasive manner?

6) Comparison of the fiber optic sensors with other sensor types used for glucose monitoring in diabetes patients should be added to the manuscript.

7) In table 1 different units are used for the same measurement; these values should be uniformed in order to facilitate any kind of comparison.

Thank you

Author Response

(The authors gave the same response as above.)

Reviewer 3 Report

Dear Authors,

I read carefully your review about fiber optic sensors for glucose measurement.

This work is interesting but must be improved.

Some sentences are difficult to understand or problematic (for example: lines 102-103, lines 167 to 171, line 222, lines 237 and 239, line 240, line 372, 391,... ).

Some data are strange (for example: line 186).

Some figures are not lisible (for example: figure 10)

Some sensitivities are not complet (for example: line 369)

Some references in the text do not correspond with Table 1 (for example: [50], [37,42, 45, 58],...)

Based on the table, what are the needs to develop a practical optical fiber glucometer (in reflexion or in transmission, in the middle or in extreme, the type of optical source,...)

There is no information about tilted fiber Bragg grating.

Regards

Author Response

(The authors gave the same response as above.)

Round 2

Reviewer 2 Report

The authors made an extraordinary effort in trying to address most of my concerns. However, a few questions remain unsolved.

When I asked what levels, in the first paragraph of the introduction, I meant the following: 

"Today, the measurement of levels, whether in humans, animals or plants."

I think the author intent is: "Today, the measurement of glucose levels, whether in humans, animals or plants".

The authors did not change this phrase in the revised manuscript.

The authors should also add a table of contents and an abbreviation list to the document.

Lastly, I emphasize the need to proofreading this manuscript.

Thank you

Author Response

(The authors gave the same response as above.)

Reviewer 3 Report

Dear Authors,

Thank you for this new version of your manuscript. A great effort has been done. Nevertheless, some sentences are strange:

1) Line 301: There are glucose detection sensors made of fiber optics by Bragg Gratings (FBG),

2) Line 323: TFBG instead of TFG 

3) Lines 329: In 2010 Saxl used a multimode mode fiber...

4) Lines: 669-676: Table 1 ?

This is only few examples. It will be nice to have a careful proofreading of the manuscript. 

Regards,

Author Response

(The authors gave the same response as above.)
